# Diverse Functions of KDM5 in Cancer: Transcriptional Repressor or Activator?

**DOI:** 10.3390/cancers14133270

**Published:** 2022-07-04

**Authors:** Yasuyo Ohguchi, Hiroto Ohguchi

**Affiliations:** Division of Disease Epigenetics, Institute of Resource Development and Analysis, Kumamoto University, Kumamoto 860-0811, Japan; urashima@kumamoto-u.ac.jp

**Keywords:** KDM5, histone demethylase, epigenetic regulator, cancer, MYC, RNA polymerase II

## Abstract

**Simple Summary:**

Aberrations in epigenetic modulators have been widely identified in cancer; therefore, these modulators are attractive targets for cancer treatment. KDM5 is an H3K4 demethylase family that is recognized as a cancer-associated epigenetic regulator. However, the role of KDM5 and its transcriptional regulation in cancer is multifaceted. Here, we provide an overview of the roles of KDM5 in cancer, explore the molecular mechanisms through which KDM5 regulates transcriptional output, including our recent finding that KDM5A supports transcriptional activation by controlling the H3K4 methylation cycle, and further discuss the possibility of the use of KDM5 inhibitors in cancer therapy.

**Abstract:**

Epigenetic modifications are crucial for chromatin remodeling and transcriptional regulation. Post-translational modifications of histones are epigenetic processes that are fine-tuned by writer and eraser enzymes, and the disorganization of these enzymes alters the cellular state, resulting in human diseases. The KDM5 family is an enzymatic family that removes di- and tri-methyl groups (me2 and me3) from lysine 4 of histone H3 (H3K4), and its dysregulation has been implicated in cancer. Although H3K4me3 is an active chromatin marker, KDM5 proteins serve as not only transcriptional repressors but also transcriptional activators in a demethylase-dependent or -independent manner in different contexts. Notably, KDM5 proteins regulate the H3K4 methylation cycle required for active transcription. Here, we review the recent findings regarding the mechanisms of transcriptional regulation mediated by KDM5 in various contexts, with a focus on cancer, and further shed light on the potential of targeting KDM5 for cancer therapy.

## 1. Introduction

Epigenetic mechanisms, including covalent modifications of DNA and histones, play important roles in cellular function and differentiation; furthermore, the dysregulation of epigenetic machinery is known to be associated with common diseases, including cancer [1,2,3]. Genetic alterations in epigenetic regulators have been widely observed in cancer cells [1,2]. Signals from the tumor microenvironment also alter the expression and function of epigenetic regulators, conferring an aberrant epigenetic landscape in cancer [3]. Therefore, cancer-related epigenetic regulators are attractive targets for cancer treatment to correct the altered epigenetic and transcriptional programs.

Histone lysine methylation is an epigenetic marker that is highly associated with gene regulation and is dynamically and reversibly modulated by two types of epigenetic regulators: lysine methyltransferases (KMTs) and lysine demethylases (KDMs) [4]. Thus, the deregulation of these enzymes has been implicated in cancer [1,2]. The roles of KMTs and KDMs other than KDM5 in cancer have been discussed in recent relevant reviews [1,2,5]. The KDM5 family is one of the Jumonji C (JmjC) domain-containing KDM families and is responsible for removing histone H3 lysine 4 di- and tri-methylation (H3K4me2 and H3K4me3), which are methyl markers associated with transcriptional activation [6,7,8,9,10]. In humans, the KDM5 family comprises four proteins: KDM5A (also known as JARID1A/RBP2), KDM5B (JARID1B/PLU-1), KDM5C (JARID1C/SMCX), and KDM5D (JARID1D/SMCY). Regardless of their structural similarity, each isoform has distinctive characteristics and functions under both physiological and pathological conditions. *KDM5A* and *KDM5B* are located on autosomal chromosomes, whereas *KDM5C* and *KDM5D* are located on the X and Y chromosomes, respectively. KDM5A and KDM5C are ubiquitously expressed in human tissues, whereas KDM5B and KDM5D are somewhat tissue-specifically expressed [11]. KDM5B is particularly detected in the testis, and the expression of KDM5D is relatively high in the small intestine [11]. KDM5A and KDM5B are crucial transcriptional regulators of the development and differentiation processes [6,7]. KDM5C is required for neural development [9], and loss-of-function mutations lead to X-linked mental retardation [12]. KDM5D is a male-specific histocompatibility antigen associated with male organ rejection in female recipients [13]. Mounting evidence indicates that KDM5A and KDM5B are oncogenic and overexpressed in cancer [14,15]. In contrast, KDM5C functions as an oncoprotein or tumor suppressor in a context-dependent manner [14,15,16]. KDM5D is recognized as a tumor suppressor [14,15,16].

In this review, we review updates in the knowledge of the diverse roles of KDM5 in cancer, highlight the recent understanding regarding the molecular mechanisms through which KDM5 is engaged in transcriptional output, including our findings that KDM5A supports MYC-driven transcription by maintaining the H3K4 methylation cycle through H3K4me3 demethylation, and further discuss the possibility of the application of KDM5 inhibitors in cancer therapy.

## 2. Structure of the KDM5 Family Proteins

The KDM5 proteins possess five conserved domains that are closely associated with their functions (Figure 1). The Jumonji N (JmjN) and JmjC domains comprise the catalytic core of this family of proteins and are essential for H3K4 demethylase activity [17]. The C_5_HC_2_ zinc finger is also required for efficient catalytic activity [17]. The AT-rich interaction (ARID) domain and the plant homeodomain (PHD) finger (PHD1) are inserted between the JmjN and JmjC domains; however, these are dispensable for catalytic activity [17]. The ARID domain recognizes a CCGCCC or GCACA/C motif, conferring DNA-binding capacity, which contributes to transcriptional regulation by KDM5 [18,19]. There are two or three PHD fingers in human KDM5 isoforms. KDM5A and KDM5B contain PHD1, PHD2, and PHD3, whereas KDM5C and KDM5D do not have PHD3. Among these PHD fingers, PHD1 is the most characterized. PHD1 recognizes the unmethylated H3K4 histone tail, thereby enhancing the demethylase activity of KDM5 [20,21,22]. Mechanistically, the binding of PHD1 to the H3 tail allosterically sustains substrate binding to the catalytic domain, thus increasing enzymatic efficiency [23,24]. PHD1 has also been shown to recognize H3K9me3 in KDM5C [9]. The function of the PHD2 finger has not yet been identified. PHD3 preferentially binds to H3K4me3 [20,25], and its binding capacity is required for NUP98–PHD fusion-related leukemogenesis [25,26,27]. While KDM5 proteins participate in diverse protein complexes to exhibit context-specific functions, these interactions mainly occur through the C-terminal part of the proteins, including PHD3 [28,29,30,31,32]. Representative binding partners of KDM5 proteins are shown in Table 1.

## 3. Transcriptional Regulation by KDM5

### 3.1. Demethylase-Dependent Transcriptional Repression

Because H3K4me3 is present at the promoters of active genes, H3K4me3 is thought to be associated with active transcription [53]. H3K4me3 is required for the selective anchoring of the basal transcription factor TFIID and for the formation of the pre-initiation complex, which is mediated by the binding of TAF3 to H3K4me3 through the PHD finger of TAF3 [54,55]. Consistent with these findings, KDM5 proteins have been shown to repress transcription via H3K4 demethylase activity in a variety of contexts, including cancer, since these proteins have been identified as H3K4 demethylases (Figure 2A) [6,7,8,9,10].

The fact that KDM5 proteins are associated with several transcriptional repressor complexes further supports the notion that KDM5 proteins are transcriptional repressors (Table 1) [20,30,33,37,39,41]. KDM5A and a Sin3 corepressor complex containing HDACs interact with each other and cooperatively repress their target genes via H3K4me3 demethylation, H3 deacetylation, and nucleosome repositioning during myogenic differentiation [37]. KDM5A also binds to the polycomb repressive complex 2 (PRC2) and mediates the transcriptional repression of polycomb target genes through the coordinated regulation of H3K4me3 demethylation and H3K27me3 methylation during embryonic stem cell differentiation [39]. In addition, KDM5A interacts with the H3K9/K27 methyltransferases G9a and GLP, and it coordinately maintains the transcriptionally repressed state of the embryonic gene E^y^ via H3K4me3 demethylation and H3K9/K27me2 methylation in adult erythroid cells [41]. Furthermore, KDM5A and KDM5B have been shown to be associated with nucleosome remodeling and the deacetylase (NuRD) complex, suggesting a collaborative repressive function of the KDM5 and NuRD complex [20,30,33]. KDM5C also forms a complex with RE1-silencing transcription factor (REST), HDAC1, HDAC2, and G9a, thereby repressing a subset of target genes of REST [10].

### 3.2. Demethylase-Independent Transcriptional Repression

KDM5 proteins have also been shown to repress transcription in a demethylase-independent manner [34,44]. Zhang et al. showed that KDM5B binds to retroelement loci and represses retroelement expression in melanoma cells [44]. However, they found that catalytic activity of KDM5B is not necessary for this function. Importantly, KDM5B recruits the H3K9 methyltransferase, SETDB1, and suppresses retroelement expression by adding H3K9me3 (Figure 2B) [44]. Ren et al. also highlighted the scaffolding function of KDM5B in acute myeloid leukemia (AML) cells [34]. KDM5B is required for recruiting the NuRD repressive complex to stemness genes to inhibit the growth of AML cells, and its demethylase activity has been proven to be dispensable for this function [34].

### 3.3. Demethylase-Dependent Transcriptional Activation

Regardless of the involvement of H3K4me3 in active transcription, KDM5 proteins paradoxically activate their target genes via H3K4 demethylase-dependent mechanisms in some contexts. Lloret-Llinares et al. examined transcriptional regulation by Little imaginal disc (Lid), which is the sole KDM5 ortholog in *Drosophila*, in the developmental genes [56]. They showed that Lid target genes are actively transcribed and enriched in Ser2/Ser5-phosphorylated RNA polymerase II (RNAPII), suggesting that Lid is a positive transcriptional regulator [56]. It was shown that the depletion of Lid downregulates its target genes, accompanied by increased H3K4me3 and reduced RNAPII phosphorylation at Ser5 at the target gene loci [56]. Based on these findings, researchers proposed that H3K4me3 demethylation by Lid is required for resetting the chromatin state and starting a new transcription cycle, the disruption of which leads to transcription failure [56]. Similar to the function of Lid in the developmental genes in *Drosophila* cells, we recently showed that KDM5A positively regulates its target genes via H3K4me3 demethylation in mammalian cells [35]. KDM5A coexists with MYC and the components of the transcriptional machinery, including CDK7, CDK9, and RNAPII, across the genome of multiple myeloma cells [35]. Importantly, KDM5A depletion and a selective catalytic inhibitor of KDM5 were found to reduce the expression of MYC target genes, concomitant with increased H3K4me3 and reduced Ser2/Ser5-phosphorylated RNAPII at the MYC target gene loci [35]. We further demonstrated that excessive levels of H3K4me3 at the transcription start site (TSS)-proximal nucleosomes lock TFIID via TAF3 anchoring, leading to hindered TFIIH and P-TEFb activities [35]. Accordingly, we proposed a model in which KDM5A supports the transcription of MYC target genes by transiently removing H3K4me3, thereby allowing for the release of TFIID and the phosphorylation of RNAPII at these gene loci (Figure 2C) [35]. KDM5A also activates the expression of pro-proliferative cell cycle genes dependent on H3K4 demethylase activity in preadipocytes [57], presumably by maintaining a proper modification cycle of H3K4 methylation.

KDM5 proteins have also been shown to activate their target genes through H3K4me3 demethylation in non-promoter regions [58,59,60,61,62]. Dahl et al. showed that ~22% of the mouse oocyte genome has broad H3K4me3 domains and that these H3K4me3 signals become restricted to TSS regions in two-cell embryos [58]. They further showed that KDM5A and KDM5B actively remove broad H3K4me3 domains and that the removal of broad H3K4me3 domains is required for the activation of normal zygotic genome and embryo development [58]. KDM5B also removes H3K4me3 in intragenic regions, which inhibits the spread of H3K4me3 into gene bodies and cryptic initiation, thereby promoting productive transcriptional elongation at the self-renewal-associated gene loci and contributing to embryonic stem cell self-renewal [59,60,61]. Finally, KDM5C has been shown to promote enhancer function by maintaining H3K4me1 through the demethylation of H3K4me3/2 [62].

### 3.4. Demethylase-Independent Transcriptional Activation

In 2000, Gildea et al. identified Lid as a member of the trithorax group that contributes to active transcription states [63]. This finding suggests that Lid acts as a transcriptional activator, and it has been shown that Lid activates transcription independently of its demethylase function [64,65,66]. Lid binds to the histone deacetylase Rpd3 and blocks its deacetylase activity, thereby upregulating the expression of the target genes of Rbp3 [64]. Lid also induces the expression of a subset of Foxo target genes by interacting with Foxo and promoting Foxo DNA binding [65]. Moreover, Lid activates genes associated with mitochondrial function through the recognition of H3K4me3 via its PHD domain [66]. Similar transcriptional activation functions of KDM5 have been described in mammalian cells. KDM5 physically binds to nuclear receptors, including the estrogen receptor (ER), and enhances their transcriptional activity (Table 1) [36,48,49,50]. KDM5A also interacts with CLOCK–BMAL1 and enhances CLOCK–BMAL1-mediated transcription by inhibiting HDAC1 function and increasing H3K9 acetylation at the *Per2* promoter (Figure 2D) [52]. Similarly, KDM5A activates metastasis-associated gene expression in breast cancer in a demethylase-independent manner [67]. Finally, KDM5B mediates the biphasic regulation of retinoic acid (RA)-dependent genes [40]. In the absence of RA, KDM5B represses RA- responsive genes by forming a complex with PRC2, whereas in the presence of RA, KDM5B activates these genes by dissociating PRC2 and recruiting AIB1-containing co-activators [40].

## 4. The Roles of KDM5 in Cancer

### 4.1. KDM5A

KDM5A was first identified as a pRb-interacting protein [42]. Later, KDM5A was shown to reinforce the repressive function of the pRb family by promoting H3K4me3 demethylation and the subsequent silencing of pRb target genes during differentiation and senescence [43,68,69]. These studies suggested that KDM5A may act as a tumor suppressor. In contrast, KDM5A has been shown to function as an oncoprotein in pRb-defective cells [70,71,72]. KDM5A promotes proliferation and blocks the differentiation of pRb-defective cells [70]. This differentiation block is mediated by the repression of mitochondrial biogenesis [70]. Accordingly, the loss of KDM5A inhibits tumorigenesis caused by the deletion of pRb in vivo [71]. Furthermore, KDM5A depletion suppresses the growth of established tumors in pRb+/− mice, indicating that KDM5A is required not only for tumor formation but also for the maintenance of established tumors driven by pRb loss [72]. The outstanding question is why KDM5A behaves in opposite ways in the presence and absence of pRb. Because KDM5A is a pRb-binding partner, the absence of pRb may alter the genome-wide distribution and function of KDM5A [73]. Further studies are required to detail the mechanism underlying the action of KDM5A in the presence or absence of pRb.

In addition to pRb, KDM5A has been shown to be associated with the function of MYC (c-Myc) [35,47]. MYC is a transcription factor implicated in oncogenesis. To identify cofactors engaged in the MYC transcriptional program in cancer, Secombe et al. performed dose-sensitive genetic screening using a *Drosophila* model and found that Lid is a positive regulator of dMyc (MYC ortholog)-induced cell growth [47]. Lid interacts with dMyc and induces the expression of the growth regulatory gene in cooperation with dMYC [47]. KDM5A and KDM5B also bind MYC, demonstrating that the KDM5–MYC connection is conserved in humans [47]. We recently showed that KDM5A upregulates MYC target genes in multiple myeloma cells [35]. KDM5A co-localizes with MYC on a genome-wide scale, and KDM5A and MYC cooperatively activate their target genes [35]. These studies suggest that KDM5A promotes MYC-driven tumorigenesis.

KDM5A has also been shown to interact with RBP-J and inhibit the expression of Notch target genes by demethylating H3K4me3 [28]. The activation of Notch signaling suppresses tumorigenesis in small cell lung cancer (SCLC) by repressing the expression of neuroendocrine genes [74], suggesting an oncogenic function of KDM5A in SCLC. Indeed, KDM5A promotes SCLC by inducing the expression of the neuroendocrine transcription factor ASCL1 through the repression of NOTCH2 and Notch target genes [75].

Fusion proteins involving NUP98 are associated with hematologic malignancies, including AML. KDM5A was identified as a fusion partner of NUP98 in 2006 [76]. Subsequent studies revealed that the NUP98–KDM5A fusion accounts for approximately 10% of pediatric non-Down syndrome-acute megakaryoblastic leukemia cases [77,78,79,80]. Multiple murine models have shown that NUP98–KDM5A fusion blocks the differentiation of hematopoietic cells and induces AML [25,81,82,83]. Mechanistically, the PHD3 finger of NUP98–KDM5A recognizes and binds to H3K4me3 at the leukemia-associated transcription factor loci, including *Hox (s)* and *Meis1*, thereby activating the transcription of these genes [25,26,27]. Consistent with these findings, tumor samples from patients with NUP98–KDM5A were characterized by the upregulation of the expression of the HOX cluster of genes [77,79]. Importantly, the abrogation of H3K4me3 recognition by mutations in the PHD finger or pharmacologic inhibition using disulfiram suppresses leukemic transformation and induces leukemia cell death [25,26,27]. Furthermore, the blockade of the menin-MLL1 interaction by the menin-MLL1 inhibitor VTP50469 hampers the chromatin binding of not only menin and MLL1 but also NUP98 fusion proteins, and it also inhibits leukemogenesis in models of NUP98-rearranged leukemias, including NUP98–KDM5A [83]. Recent studies have also shown that JAK-STAT signaling pathway genes and *CDK6* are downstream targets of NUP98–KDM5A and that NUP98–KDM5A leukemia is vulnerable to JAK or CDK6 inhibition [81,82]. Liquid–liquid phase separation plays a key role in appropriate transcription, and aberrant transcriptional condensates are associated with tumorigenesis [84]. Chandra et al. showed that NUP98 fusion proteins, including NUP98–KDM5A, form leukemia-related nuclear condensates, thus leading to leukemogenesis [85]. In addition to NUP98–KDM5A fusion, KDM5A has been shown to be involved in the progression of acute promyelocytic leukemia (APL) [86]. KDM5A contributes to the inhibition of differentiation in APL cells by repressing PML–RARα target genes via demethylating H3K4me2 [86]. Conversely, KDM5 inhibitor CPI-455 or the knockout of KDM5A sensitizes APL cells to all-trans retinoic acid-induced differentiation [86].

KDM5A has also been shown to be involved in drug resistance in cancer cells [87,88,89]. Sharma et al. found that while the non-small cell lung cancer (NSCLC)-derived cell line PC9 is sensitive to epidermal growth factor receptor tyrosine kinase inhibitors (EGFR TKIs), a small fraction of PC9 cells are tolerant to EGFR TKIs [87]. They showed that EGFR TKIs inhibit EGFR kinase activity in drug-tolerant cells, indicating that drug tolerance is not due to the enhancement of drug efflux function [87]. Furthermore, drug-tolerant cells have not acquired genetic alterations related to acquired EGFR TKI resistance, including the EGFR T790M mutation [87]. In addition, EGFR TKI-tolerant cells are also insensitive to cisplatin, suggesting that induced-drug tolerance is not pathway-specific [87]. Importantly, the drug resistance was reversed after the withdrawal of EGFR TKIs, suggesting that this phenotype is derived from epigenetic alterations [87]. Indeed, the drug-tolerant state is accompanied by chromatin alterations, and KDM5A mediates this in an H3K4 demethylase-dependent manner [87]. Later studies confirmed this notion using several drug-tolerant cell models and also showed that KDM5 inhibitors are useful for suppressing drug-tolerant cells [88,89,90].

Furthermore, KDM5A has been shown to modulate tumor immunity [91]. The overexpression of KDM5A downregulates *Pten* expression by removing H3K4me3 at the *Pten* promoter, which consequently activates the PI3K–AKT–S6K1 signaling cascade, resulting in PD-L1 accumulation and sensitization to immune checkpoint blockade in murine cancer models [91]. Consistent with these findings, higher KDM5A expression was found to be associated with a good prognosis in patients with metastatic melanoma treated with anti-PD-1 antibodies [91]. The functions of KDM5A in cancer are summarized in Figure 3.

### 4.2. KDM5B

*KDM5B* was first identified as an upregulated gene in breast cancer [92]. Later, Yamane et al. showed that KDM5B is an H3K4 demethylase that promotes the proliferation of breast cancer cells by repressing tumor suppressor genes, including *HOXA5* and *BRCA1*, through H3K4me3 demethylation [8]. Subsequent studies further delineated the functions of KDM5B in breast cancer. *KDM5B* is frequently amplified and overexpressed in ER^+^ luminal breast cancer cells, and KDM5B governs the luminal lineage transcriptional program [93]. Higher KDM5B activity is associated with poor prognosis in patients with ER^+^ breast cancer [93]. *EMSY* is also amplified in ER^+^ luminal breast cancer and acts as an oncogene by downregulating miR-31, an antimetastatic microRNA [38]. KDM5B physically interacts with EMSY and suppresses miR-31 via H3K4me3 demethylation [38]. KDM5B also interacts with ER and enhances E2-dependent tumor growth in ER^+^ breast cancer cells [49]. Mitra et al. showed that KDM5B regulates cyclinD1 expression and cell cycle progression by suppressing let-7e tumor suppressor microRNA in ER^+^ breast cancer cell lines [94]. Furthermore, KDM5B confers a transcriptomic heterogeneity in ER^+^ breast cancer, and high KDM5B expression increases the risk of therapeutic resistance [95]. This evidence supports the notion that KDM5B is oncogenic in ER^+^ breast cancer. In contrast, KDM5B may function as a tumor suppressor in triple-negative breast cancer (TNBC). The expression of KDM5B is relatively lower in TNBC than in ER^+^ breast cancer [20]. This observation is consistent with the finding that KDM5B regulates luminal cell-specific expression programs [93] and suggests that KDM5B may be dispensable for TNBC. Importantly, the overexpression of KDM5B reduces the migratory and invasive activities of TNBC cell lines, whereas the knockdown of KDM5B increases these activities [20,30,96]. The overexpression of KDM5B also results in a reduction in cancer stem cell frequency in TNBC cell lines [96], suggesting tumor-suppressive roles of KDM5B in TNBC. However, other groups have shown that KDM5B promotes metastasis and progression in TNBC [97,98]. Further studies are required to confirm the definitive roles of KDM5B in TNBC.

In melanoma, KDM5B expression progressively decreases with intratumoral heterogeneity [99]. KDM5B is highly expressed in benign melanocytic nevi, whereas only a small fraction of tumor cells are KDM5B-positive in advanced and metastatic melanomas [99]. Based on these findings, early studies suggested the tumor-suppressive roles of KDM5B in melanoma [100,101]; however, later studies revealed its oncogenic functions in melanoma [102,103,104]. Roesch et al. showed that KDM5B is highly expressed in a small population of slow-cycling melanoma cells and that KDM5B knockdown initially accelerates melanoma cell growth followed by exhaustion, indicating that KDM5B-positive cells have cancer stem cell-like features [102]. These KDM5B^high^ subpopulations also contribute to intrinsic multidrug resistance by promoting oxidative phosphorylation and maintaining a high ratio of reduced-to-oxidized glutathione [103,104]. On the other hand, enforced KDM5B expression limits tumor plasticity and heterogeneity, resulting in the elimination of melanoma cells by promoting melanocytic differentiation [105]. The authors of a recent study identified an immune evasion function driven by KDM5B in melanoma [44]. KDM5B expression is significantly higher in tumors from non-responders to immune checkpoint blockade than in those from responders [44], and KDM5B inhibits anti-melanoma immunity by recruiting the H3K9 methyltransferase SETDB1 to suppress endogenous retroelements in a demethylase-independent fashion [44]. Conversely, KDM5B depletion activates these retroelements, leading to a type I interferon response and tumor rejection [44].

KDM5B expression is higher in prostate cancer than in normal or benign tissues, and higher KDM5B expression is related to advanced tumors and poor prognosis [50,106]. Xiang et al. showed that KDM5B physically binds to androgen receptors (ARs) and enhances AR transcriptional activity in an enzymatic-dependent manner [50]. KDM5B also plays a key role in activating PI3K/AKT signaling in prostate cancer [107]. The knockout of *Kdm5b* was found to suppress prostate tumorigenesis in a *Pten*-null prostate cancer mouse model by reducing the levels of P110α/P85 [107]. KDM5B depletion also reduces P110α and PIP3 levels and subsequently inhibits the proliferation of human prostate cancer cells [107]. Interestingly, AKT has been shown to positively regulate the expression of KDM5B by repressing miR-137 in PTEN-null prostate cancer [108], suggesting that KDM5B forms a feed-forward regulatory loop with PI3K/AKT signaling in prostate cancer.

KDM5B is also implicated in NSCLC. The expression of KDM5B is elevated in NSCLC tissues and is negatively associated with overall survival in NSCLC patients [109]. KDM5B promotes epithelial–mesenchymal transition by increasing the expression levels of ZEB1 and ZEB2 by repressing the miR-200 family [110]. KDM5B also induces a cancer stem cell-like phenotype by activating the c-MET signaling pathway in NSCLC cells [110]. Furthermore, KDM5B confers radioresistance to NSCLC cells [111,112]. Bayo et al. showed that at sites of DNA damage, H3K4me3 interferes with the recruitment of DNA repair factors, and that in irradiated cancer cells, KDM5B facilitates DNA repair by removing H3K4me3 at those sites [111]. Conversely, KDM5B inhibition impairs DNA repair, leading to the sensitization of cancer cells to radiation [111].

KDM5B has also been reported as an oncogene in other types of cancers, including hepatocellular carcinoma, esophageal cancer, gastric cancer, colorectal cancer, oral cancer, Ewing sarcoma, glioma, acute lymphoblastic leukemia, and chronic myeloid leukemia [32,113,114,115,116,117,118,119,120,121,122,123,124,125,126,127,128,129]. Overall, KDM5B plays a key role in cancer progression by promoting cancer stemness, epithelial–mesenchymal transition, the cell cycle, DNA repair, and intratumoral heterogeneity.

In contrast, KDM5B serves as a tumor suppressor in AML with mixed-lineage leukemia (MLL) rearrangement or NUP98–NSD1 [34,130]. Wong et al. showed that global H3K4me3 levels are higher in leukemia stem cells (LSCs) than in differentiated leukemia cells in MLL-rearranged AML, as well as that the expression of LSC maintenance genes is reduced in association with decreased H3K4me3 levels and LSC differentiation [130]. Importantly, KDM5B is an H3K4 demethylase that is induced in differentiated leukemia cells, and the overexpression of KDM5B abrogates the oncogenic potential of LSC in MLL-rearranged AML [130]. Ren et al. also showed that KDM5B suppresses LSC-related genes and AML progression in AML with MLL rearrangement or NUP98–NSD1 [34]. However, they revealed that this function is independent of the demethylase activity of KDM5B, suggesting a scaffold function of KDM5B to recruit NuRD repressive complex [34]. The functions and roles of KDM5B in each type of cancer are summarized in Figure 4 and Table 2.

### 4.3. KDM5C

Loss-of-function mutations in *KDM5C* have been identified in several types of cancers, including clear cell renal cell carcinoma (ccRCC), and *KDM5C* is thought to be a tumor suppressor gene [131,132]. It has been shown that the overexpression of KDM5C suppresses tumorigenesis, whereas the depletion of KDM5C has been shown to enhance tumor growth in vitro and in mouse xenograft models [133,134,135]. KDM5C inhibits the super-enhancer activity of oncogenes such as S100A, EGFR, and c-MET by repressing enhancer RNA transcription [46,134]. KDM5C also suppresses human papillomavirus E6 and E7 oncogene expression as an E2-mediated co-repressor [136]. Interestingly, *KDM5C* has been shown to be one of the X-chromosome genes that escapes from X-inactivation and to be more frequently mutated in males, suggesting that sustained KDM5C expression is part of the reason for the reduced cancer incidence in females [132].

So far, the tumor-suppressive roles of KDM5C have been most studied in ccRCC (Figure 5). KDM5C deficiency leads to the abnormal expression of heterochromatic non-coding RNAs, thereby increasing the genomic instability in ccRCC [137]. The integrated multi-omics analysis of RCC samples has revealed that loss-of-function mutations in *KDM5C* are associated with angiogenic features [138]. Importantly, most ccRCC tumors with *KDM5C* mutations have *VHL* mutations, suggesting that the inactivation of both genes enhances tumorigenesis [131]. Indeed, KDM5C knockdown was found to increase the tumor size of VHL-deficient RCC cells in a xenograft model [133]. A mechanistic study showed that, after VHL inactivation, HIF upregulates the tumor suppressor ISGF3, which is suppressed by KDM5C inactivation [139].

KDM5C also inhibits fatty acid metabolism by reducing FASN expression in intrahepatic cholangiocarcinoma [135]. In precursor B cell leukemia harboring *ETV6*–*RUNX1* fusion, KDM5C deficiency promotes aberrant RAG recruitment to elevated H3K4me3 sites, conferring RAG off-target cleavage activity and the clonal evolution of leukemia cells [140].

In contrast, *KDM5C* has also been shown to be overexpressed and to function as an oncogene in some cancers. KDM5C facilitates castration-resistant prostate cancer cell growth by downregulating PTEN [141]. It promotes breast cancer tumorigenesis by activating ERα-target genes or repressing BRMS1 expression and protects breast cancer cells from immune surveillance by repressing type I interferons and interferon-stimulated genes [36,142]. KDM5C also plays oncogenic roles in gastric cancer, colon cancer, hepatocellular carcinoma, and lung cancer by suppressing p53, FBXW7, BMP7, and miR-133a, respectively [143,144,145,146].

### 4.4. KDM5D

*KDM5D*, encoded on the Y chromosome, has recently been shown to be a tumor suppressor gene. The incidence of ccRCC and lung cancer is higher in men than in women, and men have poorer outcomes [147,148]. Importantly, the total or partial loss of the Y chromosome, including the *KDM5D* locus, has been observed not only in prostate cancer but also in ccRCC and lung cancer in males, and lower KDM5D expression is associated with poor prognosis [147,148,149,150]. Decreased KDM5D expression has also been observed in gastric and colorectal cancers [151,152]. In prostate cancer, KDM5D suppresses the invasive phenotype by reducing the expression of metastasis-associated genes, such as MMPs and Slug, by interacting with ZMYND8 and removing H3K4me3 marks [45,149]. KDM5D also represses the cell cycle and mitotic entry, and the loss of KDM5D accelerates the cell cycle, resulting in DNA-replication stress with ATR activation [150]. Thus, the ATR inhibitor is sensitive to *KDM5D*-defective prostate cancer [150]. Furthermore, KDM5D directly interacts with AR and represses the target genes of AR by demethylating H3K4me3, and KDM5D depletion enhances the transcriptional activity of AR, leading to docetaxel insensitivity [51]. Indeed, KDM5D expression predicts the response to docetaxel in patients with metastatic castration-resistant prostate cancer [153]. KDM5D has also been shown to inhibit the invasion and metastasis of gastric and colorectal cancer cells [151,152], as well as to prevent DNA damage in hematopoietic stem and progenitor cells and leukemogenesis [154].

## 5. Potent KDM5-Selective Inhibitors

As discussed above, KDM5 plays an oncogenic role in various types of cancers. Therefore, the development of KDM5 inhibitors is of particular interest. Although it is important to develop KDM5-selective inhibitors to reduce unfavorable adverse effects, this is challenging because of the structural similarity between the catalytic domains of JmjC domain-containing proteins. Despite this obstacle, prototypes of KDM5-selective inhibitors have been recently identified. KDM5-C49, a 2,4-pyridinedicarboxylic acid analog, has been shown to possess a high degree of selectivity for KDM5 demethylases in biochemical assays [155], and KDM5-C70 (an ethyl ester pro-drug of KDM5-C49) and KDOAM-25 (a corresponding amide analog of KDM5-C70) have been shown to specifically increase H3K4me3 at the cellular level [95,155,156]. Consistent with the oncogenic roles of KDM5 in cancers, KDM5-C70 and KDOAM-25 inhibit the growth of MM.1S myeloma cells and luminal breast cancer cells [95,155,156]. However, these compounds exert only modest cellular effects due to their poor cellular permeability. To overcome this problem, we developed JQKD82, a phenol ester derivative of KDM5-C49 [35]. JQKD82 increases intracellular compound concentrations and H3K4me3 levels in cells, and it better inhibits the growth of MM.1S cells compared to KDM5-C70 [35]. Importantly, JQKD82 was shown to suppress the growth of myeloma cells, not only in vitro but also in vivo in mouse xenograft models without apparent toxicity, suggesting a promising therapeutic index for targeting KDM5 [35]. Thus, JQKD82 has proven to be a useful tool for investigating the in vitro and in vivo functions of KDM5. CPI-455, a pyrimidine derivative, was also shown to selectively inhibit KDM5 demethylase activity in biochemical and cellular contexts [88]. CPI-455 demonstrated ~200-fold selectivity for KDM5A over KDM4C, which is structurally similar to the KDM5 family proteins [88]. CPI-455 reduces the number of drug-tolerant cancer cells, as well as the growth of breast cancer cells in vitro; however, the in vivo efficacy of this compound has not been determined [88,157]. Other potent and promising KDM5 inhibitors have also been developed, although further validations are required to ensure selectivity toward KDM5 enzymes and the cellular effects of these compounds [158,159].

## 6. Conclusions and Perspectives

Accumulating evidence indicates an essential role of KDM5 in cancer. Although each KDM5 isoform has diverse functions in different contexts, KDM5A and KDM5B mainly contribute to tumorigenesis by blocking differentiation, promoting cancer cell proliferation and invasion, and inducing cancer stemness and drug resistance. Previous studies have shown that these functions are mediated by transcriptional repression via H3K4 demethylase activity. However, as discussed in this review, recent studies have revealed that these functions are mediated by multiple transcriptional regulation in a context-dependent manner: transcriptional repression or activation in an H3K4 demethylase-dependent or -independent manner. These mechanistic insights provide guidance for the development of novel therapeutic strategies for each type of cancer. Inhibitors targeting the enzymatic activity of KDM5 are promising for treating demethylase-dependent cancers. Notably, prototypes of KDM5-selective inhibitors have already been developed, and preclinical studies using these compounds have shown encouraging results. However, the development of KDM5 protein degraders or protein–protein interaction inhibitors for KDM5-associated complexes is required for treating cancers that depend on the non-demethylase functions of KDM5. In addition to efforts to develop clinical-grade KDM5 inhibitors and degraders, further mechanistic studies are required for the development of rational combination therapies with KDM5 inhibitors because the cancer inhibitory effect of KDM5 inhibitors alone has been shown to be durable but relatively mild in preclinical studies. These mechanistic insights can be leveraged to develop novel epigenetic therapies to improve the outcomes of cancer patients. 

## Figures and Tables

**Figure 1 cancers-14-03270-f001:**
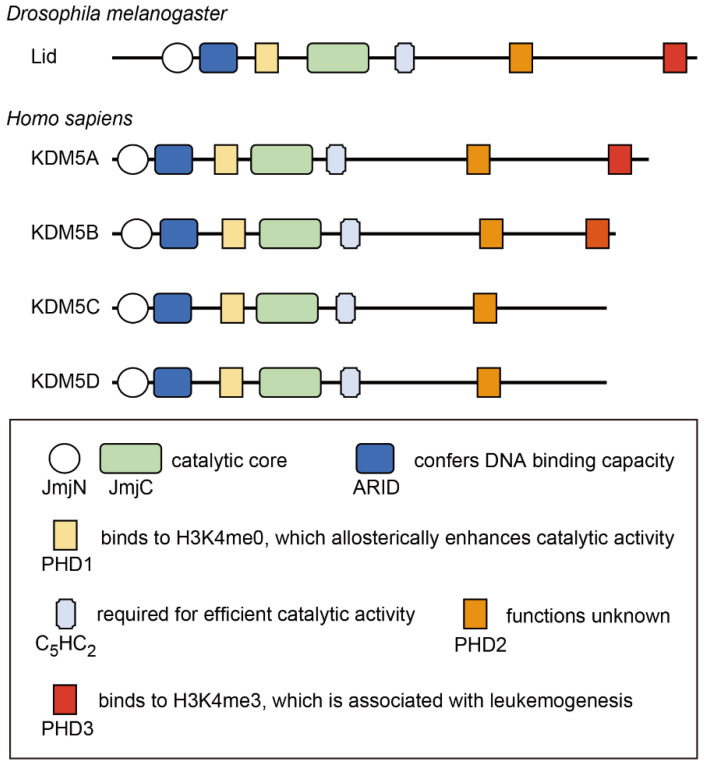
Domain structure of the KDM5 protein family. The functions of each domain are also presented. JmjN: Jumonji N domain; JmjC: Jumonji C domain; ARID: AT-rich interaction domain; PHD: plant homeodomain; C_2_HC_2_: C_5_HC_2_ zinc finger.

**Figure 2 cancers-14-03270-f002:**
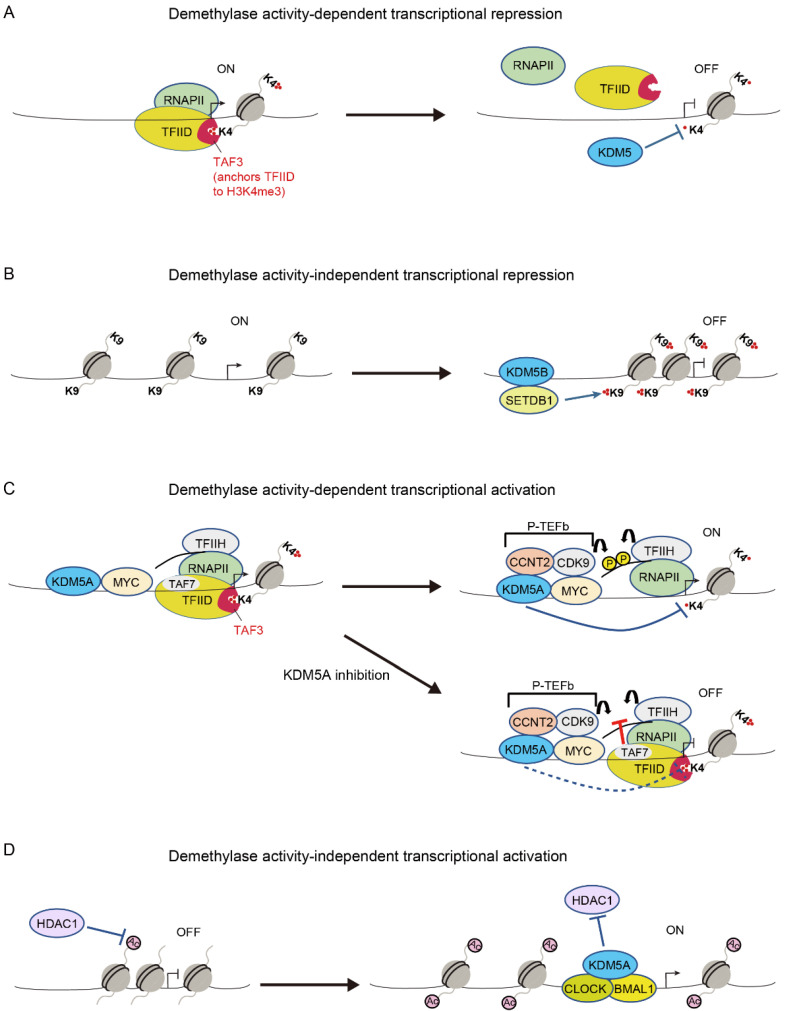
Transcriptional regulation by KDM5. (**A**) Pre-initiation complex assembly is mediated by the selective anchoring of the basal transcription factor TFIID to the core promoter via TAF3 binding to H3K4me3. KDM5 proteins remove H3K4me3, thereby preventing transcription. (**B**) KDM5B functions as a scaffold protein and recruits the H3K9 methyltransferase, SETDB1, thereby suppressing retroelement expression by adding H3K9me3 [44]. (**C**) After pre-initiation complex formation, KDM5A supports the transcription of MYC target genes via the timely removal of H3K4me3, thereby allowing for the release of TFIID and for the phosphorylation of RNA polymerase II (RNAPII). KDM5A inhibition blocks H3K4me3 demethylation, thereby locking TFIID via TAF3 anchoring and impairing the activities of TFIIH and P-TEFb, which may be mediated by TAF7 [35]. (**D**) KDM5A promotes CLOCK–BMAL1-mediated transcription by inhibiting HDAC1 [52].

**Figure 3 cancers-14-03270-f003:**
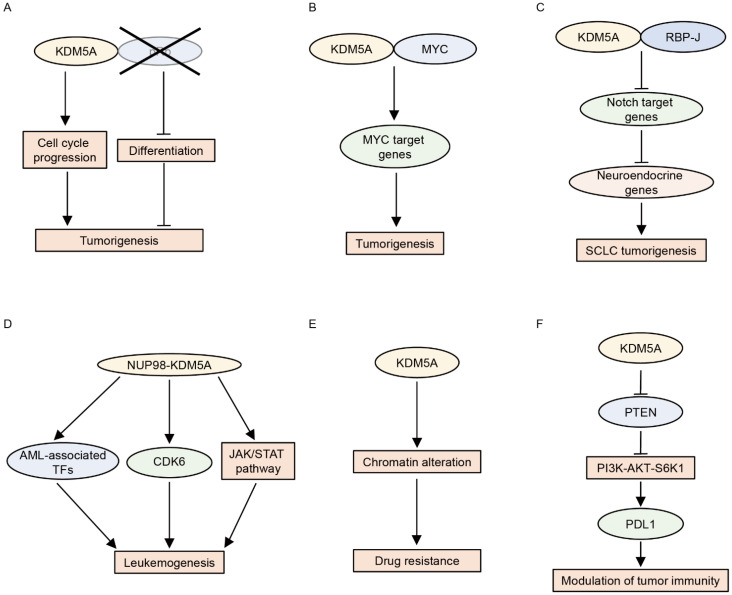
Multiple functions of KDM5A in cancer. (**A**) In pRb-deficient cells, KDM5A promotes tumorigenesis by cell cycle progression and differentiation block. (**B**) KDM5A and MYC coordinately promote tumorigenesis by activating MYC target gene expression. (**C**) KDM5A and RBP-J promote small cell lung cancer (SCLC) by inducing the expression of neuroendocrine genes through the repression of Notch target genes. (**D**) NUP98–KDM5A induces leukemogenesis by activating the expression of acute myeloid leukemia (AML)-associated transcription factors (TFs) and CDK6, as well as the JAK-STAT pathway. (**E**) KDM5A confers drug resistance by chromatin alterations. (**F**) KDM5A elevates PDL1 expression by activating the PI3K–AKT–S6K1 cascade through the downregulation of PTEN.

**Figure 4 cancers-14-03270-f004:**
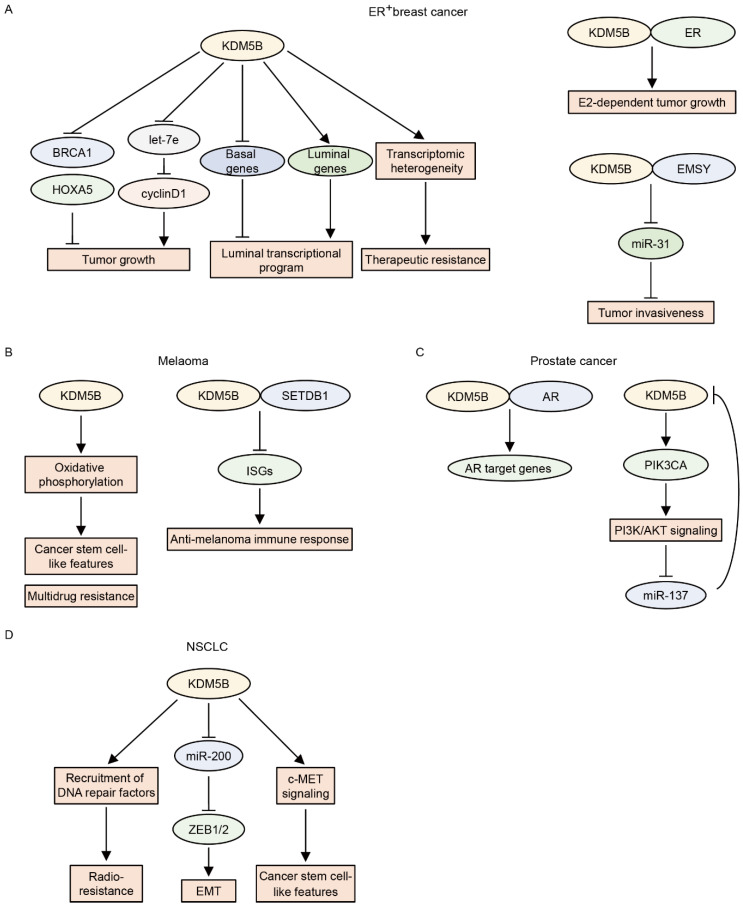
The functions of KDM5B in each type of cancer. (**A**) In estrogen receptor (ER)^+^ breast cancer, KDM5B promotes tumor growth by repressing BRCA1 and HOXA5 and by inducing cyclin D1 through the repression of let-7e. KDM5B induces the luminal transcriptional program by downregulating basal genes and upregulating luminal genes. KDM5B also contributes to therapeutic resistance by inducing transcriptomic heterogeneity. KDM5B and ER coordinately stimulate E2-dependent tumor growth. KDM5B and EMSY induce tumor invasiveness by repressing miR-31. (**B**) In melanoma, KDM5B induces cancer stem cell-like features and multidrug resistance by promoting oxidative phosphorylation. KDM5B also inhibits the anti-melanoma immune response by repressing interferon-stimulated genes (ISGs) in cooperation with SETDB1. (**C**) In prostate cancer, KDM5B activates androgen receptor (AR) target genes in cooperation with AR. KDM5B also activates PI3K/AKT signaling by upregulating PIK3CA. PI3K/AKT signaling suppresses miR-137, which consequently increases KDM5B expression, forming a positive feedback loop. (**D**) In non-small cell lung cancer (NSCLC), KDM5B induces epithelial–mesenchymal transition (EMT) by increasing ZEB1 and ZEB2 through repressing the miR-200. KDM5B also confers radioresistance by recruiting DNA repair factors to DNA damage sites. In addition, KDM5B induces cancer stem cell-like features by activating c-MET signaling.

**Figure 5 cancers-14-03270-f005:**
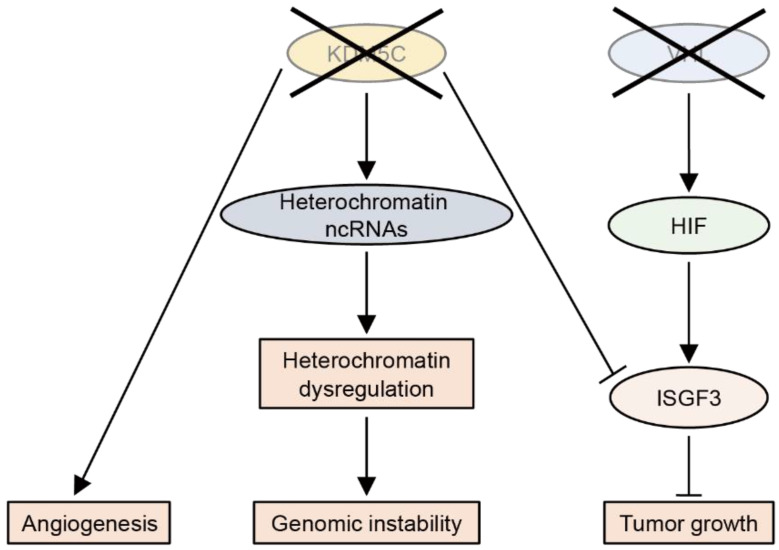
The loss-of-function of KDM5C promotes clear cell renal cell carcinoma (ccRCC). The loss-of-function of KDM5C increases genomic instability in ccRCC by heterochromatin dysregulation through the upregulation of non-coding (nc)RNAs. KDM5C deficiency also promotes angiogenesis. In VHL-deficient ccRCC cells, KDM5C deficiency promotes tumor growth by suppressing the tumor suppressor ISGF3, which is induced by HIF.

**Table 1 cancers-14-03270-t001:** Representative binding partners of KDM5 proteins.

Partner Proteins	KDM5 Isoforms	Functions	References
HDAC1/HDAC2	KDM5A/KDM5B/KDM5C	Transcriptional repression	[10,20,30,33,34,35,36]
LSD1	KDM5B	Transcriptional repression	[30]
SIN3B	KDM5A	Transcriptional repression	[33,35,37]
EMSY	KDM5A/KDM5B	Transcriptional repression	[33,35,38]
EZH2/SUZ12	KDM5A/KDM5B	Transcriptional repression	[39,40]
G9a	KDM5A/KDM5C	Transcriptional repression	[10,41]
REST	KDM5C	Transcriptional repression	[10]
RBP-J	KDM5A	Transcriptional repression	[28]
pRb	KDM5A	Transcriptional repression	[42,43]
KLF10	KDM5B	Transcriptional repression	[29]
SETDB1	KDM5B	Transcriptional repression	[44]
ZMYND8	KDM5A/KDM5C/KDM5D	Transcriptional repression or activation	[33,35,36,45,46]
MYC	KDM5A/KDM5B	Transcriptional repression or activation	[31,35,47]
CCNT2/CDK9	KDM5A	Transcriptional activation	[35]
CCNT1/CDK9	KDM5C	Transcriptional activation	[36]
ER	KDM5A/KDM5B/KDM5C	Transcriptional activation	[36,48,49]
RAR	KDM5A/KDM5B	Transcriptional repression or activation	[40,48]
AR	KDM5B/KDM5D	Transcriptional repression or activation	[50,51]
CLOCK/BMAL1	KDM5A	Transcriptional activation	[52]

ER: estrogen receptor; RAR: retinoic acid receptor; AR: androgen receptor.

**Table 2 cancers-14-03270-t002:** The role of KDM5B in each type of cancer.

Type of Cancer	Role	Prognosis	References
ER^+^ breast cancer	Oncogene	Poor in patients with high KDM5B expression	[8,38,49,93,94,95]
Triple-negative breast cancer	Tumor suppressor or Oncogene?	Poor in patients with high KDM5B expression	[20,30,96,97]
Melanoma	Oncogene	Poor response to ICB in patients with high KDM5B expression	[44,102,103]
Prostate cancer	Oncogene	Poor in patients with high KDM5B expression	[50,106,107,108]
Non-small cell lung cancer	Oncogene	Poor in patients with high KDM5B expression	[109,110,111,112]
Hepatocellular carcinoma	Oncogene	Poor in patients with high KDM5B expression	[113,120,121]
Esophageal cancer	Oncogene	Poor trend in patients with high KDM5B expression	[119,122,123]
Gastric cancer	Oncogene	NA	[114]
Colorectal cancer	Oncogene	Poor in patients with high KDM5B expression	[115,124,125]
Oral cancer	Oncogene	Poor in patients with high KDM5B expression	[116,126]
Ewing sarcoma	Oncogene	Poor in patients with high KDM5B expression	[118]
Glioma	Oncogene	Poor in patients with high KDM5B expression	[127,128]
ALL	Oncogene	NA	[117,129]
CML	Oncogene	NA	[32]
AML with MLL rearrangement or NUP98–NSD1	Tumor suppressor	Good in patients with high KDM5B expression	[34,130]

ER: estrogen receptor; ICB: immune checkpoint blockade; NA: not available; ALL: acute lymphoblastic leukemia; CML: chronic myeloid leukemia; AML: acute myeloid leukemia.

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
