# Peer review of "Diverse Functions of KDM5 in Cancer: Transcriptional Repressor or Activator?"

_cancers, 2022, doi:10.3390/cancers14133270_

Round 1

Reviewer 1 Report

This review of the KDM5 genes/ proteins  gives a fair review of the changes that can occur in Cancer.There are some points regarding the omission of  of some very recent references or the discussion of references to other KDM5 proteins - particularly KDM5B   as the authors refer dominantly to the KDM5A homologue. They need to modify the manuscript to respond  to these points.  

Reviewer 2 Report

The Authors in the MS „Diverse functions of KDM5 in cancer: transcriptional repressor or activator?” describe the function of lysine demethylases emphasizing their role in transcriptional activation and role in cancer progression. The article is well structured (divided into 6 parts) and the Introduction part explains the basics and shows the differences between KDM family members. However, the paper required significant English improvement and reorganising text to improve the flow of MS.

Part 3 is hard to follow. I would recommend reorganizing this chapter. However, the panel describing transcription elongation is presented in an interesting way but the Authors should more precisely use terms initiation and elongation of transcription as these are not interchangeable. If mechanistic details are unclear word “transcription” can be used instead (Fig. 2).

In part 4 the Authors describe the role of single members of the KDM5 family in cancer. The MS will benefit from the more story-telling narration about KDM5A than description as a collection of facts. Additionally, I would recommend compressing information from ref 43 as it is hard to follow in its current form. A similar observation can be made for ref 53. The take-home message is not clear.

The drug-related section could be better described. The authors mention cell tolerance (KDM5A inhibitors). I would advise giving an opinion (based on literature) about the questions: (1) Do cells harbour classic drug resistance driver alterations after drug treatment? (2) Is their partial resistance phenotype transient and reversible upon removal of the drug?

FTo make it easier for Readers I would suggest preparing signalling panels based on the text instead of Figure 3.

The KDM5B part is well-written. I advise preparing a table with the patient-based results described here (how KDM5B works in terms of BC type, and different types of cancers, patient prognosis and so on) and signalling panels (as for KDM5A).

„KDM5B has 302 also been reported to promote metastasis and progression in triple-negative breast cancer 303 (TNBC)[93,94]; however, these results remain controversial”. This could be explained a bit deeper. The review is a good opportunity to discuss different results.

In the KDM5C part I would recommend describing a specific type of cancer in one paragraph (ccRCC) and adding panels.

In general: Table (patient outcomes) and signalling panels are needed. Additionally, some proteins which interact with KDM5 and others are not introduced (for example RARα, estrogen receptor, EMSY) which makes it difficult to track.

Please, check carefully when using H3K4 and H3K4Me3. (For example, „…through H3K4 demethylation”)

The inhibitors part could be better discussed (important part taking into account that KDMs could play as tumor promotor) and some additional positions could be added: doi: 10.1016/j.chembiol.2017.02.006, doi: 10.3390/molecules24091739 and more.

Reviewer 3 Report

The authors aim at reviewing the current knowledge on the KDM5 lysine demethylases and their role in cancer. The review is very informative, well structured and quite comprehensive. I recommend this manuscript for publication in Cancers. However, I suggest that some minor issues are addressed:

line #89: Instead of “through the C-terminal of the proteins” it should probably say “through the C-terminus of the proteins” or “through the C-terminal part of the proteins”.

line #109: Instead of “H4K3 demethylation” it should probably say “H3K4 demethylation”
